# BOOSTING ADVERSE WEATHER CROWD COUNTING VIA MULTI-QUEUE CONTRASTIVE LEARNING

## ABSTRACT

Currently, most crowd counting methods have outstanding performance under normal weather conditions. However, they often struggle to maintain their performance in extreme and adverse weather conditions due to significant differences in the domain and a lack of adverse weather images for training. To address this issue and enhance the model's robustness in adverse weather, we propose a two-stage crowd counting method. In the first stage, we introduce a multi-queue MoCo contrastive learning strategy to tackle the problem of weather class imbalance. This strategy facilitates the learning of weather-aware representations by the model. In the second stage, we employ the supervised contrastive loss to guide the refinement process, enabling the conversion of the weather-aware representations to the normal weather domain. In addition, we also create a new synthetic adverse weather dataset. Extensive experimental results show that our method achieves competitive performance.

## 1 INTRODUCTION

Crowd counting has attracted much attention recently due to its wide range of applications such as public safety, video surveillance, and traffic control. Currently, most of the crowd counting methods (Zhang et al., 2016; Li et al., 2018; Lin et al., 2022) are able to estimate the number of crowds well on the images recorded under normal weather conditions. However, when it comes to adverse weather conditions such as rain, haze, and snow, these methods face challenges in maintaining their performance due to domain differences and the limited availability of adverse weather images (as depicted in Fig. 1(c)).

To mitigate the negative influence of adverse weather, an intuitive remedy is to pre-process the images using image restoration modules before counting. Unfortunately, even when restoration modules can mitigate the obscuration caused by adverse weather, the restored images still exhibit significant domain differences from normal weather images. Moreover, the additionally introduced classification and enhancement modules significantly increase the computational burden. Huang

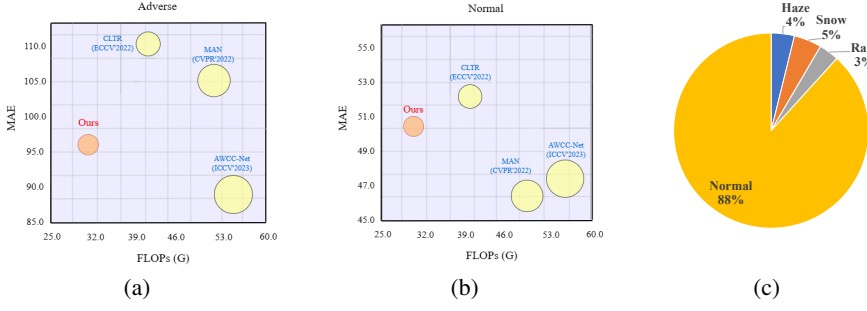

Figure 1: Trade-off of model weight and accuracy between our proposed method and state-of-the-art methods under adverse weather (a) and normal weather (b) conditions on the JHU-Crowd++ dataset. The radius of the circle is proportional to the number of parameters of the model. Under adverse conditions, the MAE performance of the state-of-the-art methods degrades by an average of 105.9%. The proportion of different weather in JHU-Crowd++ is shown in (c).

et al. (2023) introduced a transformer-based approach that addresses the issue of generating complementary information from image-specific degradation without the need for customized image enhancement modules. However, this transformer-based approach not only poses a heavy computational burden but also ignores the issue of weather class imbalance in the dataset.

The aim of this study is to enhance the robustness of crowd counting models in unknown adverse weather conditions while maintaining their performance under normal weather conditions without significantly increasing model complexity. To achieve this objective, we approach the problem of crowd counting, which includes both normal and adverse weather conditions, as an imbalanced multi-domain learning task. The key prerequisite for a model to tackle multi-domain learning is to have perception capabilities across different domains, meaning that the extracted information from each domain should possess discriminative characteristics. This aligns with the objective of contrastive learning (Chen et al., 2020; He et al., 2020; Oord et al., 2018). Therefore, in this paper, we propose a two-stage method called **M**ulti-**q**ueue **C**ontrastive **L**earning (**MQCL**). This approach enables the backbone model to directly extract weather-aware representations, which are further refined by a refiner module. In the first stage, we use unsupervised contrastive learning to distinguish the characteristics of different weather. However, since the class imbalance mentioned above, the gradient of the loss function of the vanilla contrastive method would be dominated by normal weather images, resulting in poor performance of representations and difficulty in refining and counting. To tackle such an imbalance problem, we design a simple yet effective contrastive learning method called multi-queue MoCo, which replaces the standard single queue in MoCo (He et al., 2020) with multiple queues, providing class-balanced key vectors. In the second stage, the supervised contrastive learning method (Khosla et al., 2020) is used to guide the refiner to convert the representations of adverse weather images to the domain of normal images. Benefiting from the effective representation learning, the refiner and the decoder can be designed to be light to meet our target of not significantly increasing the weight. Compared to the backbone model ConvNeXt (Liu et al., 2022; Ling et al., 2023) used in this paper, our method only introduces **15.3%** of the extra FLOPs and **12.7%** parameters. Comparison of weight and accuracy of MQCL and state-of-the-art methods is shown in Fig. 1(a/b).

In addition, as currently only one publicly available dataset, JHU-Crowd++ (Sindagi et al., 2020), contains adverse weather images and annotations, we synthesized a new dataset called NWPU-Weather based on the NWPU-Crowd dataset (Wang et al., 2020b) with rainy and hazy scenes. This dataset aims to facilitate research on crowd counting in adverse weather conditions. Several representative counting networks are benchmarked to provide an overview of the state-of-the-art performance. Codes and the NWPU-Weather dataset are available at: `https://anonymous.4open.science/r/MQCL-B46E/`.

The main contributions of our paper are concluded as follows.

- To boost the robustness of the model under adverse weather conditions while maintaining normal-weather performance, we propose a lightweight two-stage method, achieving significant improvement compared to the baseline.
- To tackle the problem of class imbalance in contrastive learning, we propose a new method called multi-queue MoCo, achieving better performance than vanilla single-queue MoCo.
- To realize the conversion of representations from the adverse weather domain to the normal weather domain, we propose a refining module guided by supervised contrastive learning, enabling the decoder to focus on a single domain.
- To provide the crowd counting field with more experimental samples in adverse weather, we synthesize a new adverse weather crowd counting dataset called NWPU-Weather. Extensive experimental results show that our method achieves competitive results.

## 2 RELATED WORK

### 2.1 CROWD COUNTING UNDER NORMAL CONDITIONS

Up to now, most single image crowd counting methods can be divided into two categories: regression-based and detection-based crowd counting. Regression-based methods mostly aim to generate a density map, and the sum of pixel values of which is the estimated total number. MCNN (Zhang et al., 2016) is a pioneer in employing such a method. Benefiting from the multi-column design, MCNN can handle input images of arbitrary size or resolution. CSRNet (Li et al.,

2018) employs dilated CNN for the back end to deliver larger reception fields and to avoid pooling operations. More recently, Lin et al. (2022) proposed a multifaceted attention network to improve transformer models in local spatial relation encoding. Du et al. (2023) redesigned the multi-scale neural network by introducing a hierarchical mixture of density experts.

In addition to model architecture, loss function designing (Ma et al., 2019; Wan et al., 2021) is also a focused area of regression-based crowd counting, which enables the models to effectively learn from ground truth.

## 2.2 Crowd Counting under Adverse Conditions

Existing deep-learning-based methods have achieved unprecedented success with crowd counting, but their performance degraded severely under adverse conditions (e.g., adverse weather) due to the disturbance to the brightness and gradient consistency. However, few research efforts have been made into this problem. Additional class conditioning blocks are utilized by Sindagi et al. (2020) to augment the backbone module, which is trained via cross-entropy error using labels available in the dataset. Huang et al. (2023) enabled the model to extract weather information according to the degradation via learning adaptive query vectors, but the weight of the model is significantly increased due to the introduction of a Transformer-based module. Kong et al. (2023) proposed a single-stage hazy-weather crowd counting method based on direction-aware attention aggregation. However, their method only focuses on the performance in hazy scenes and cannot handle various unknown weather conditions.

## 2.3 Contrastive Learning

Contrastive learning (Oord et al., 2018; Chen et al., 2020) has attracted much attention due to its success in unsupervised representation learning. The target of it is to maximize the similarity of the representations between positive pairs while minimizing that of negative pairs. In recent years, there has been a lot of work to tap the potential of contrastive learning. For example, He et al. (2020) built a dynamic dictionary with a queue and a moving-averaged encoder to enable large-scale contrastive learning with dramatically low demand for memory. Khosla et al. (2020) investigated the contrastive loss and adapted contrastive learning to the field of supervised learning.

## 3 Proposed Method

In this work, we aim to improve the robustness of the model under multiple adverse weather conditions and maintain good performance under normal weather. Each sample in the training set consists of three components: the input image, the ground truth points of the human heads and a class label representing the class of the weather. Noting that the weather label is not available to the model in the inferencing phase, which requires the model to be able to deal with unknown corruptions. Images under adverse weather represent only a small part of the dataset. Thus, we formulate our problem as an imbalanced multi-domain learning problem.

## 3.1 Framework Overview

The architecture of our method is illustrated in Fig. 2. As discussed earlier, we aim to directly enable the crowd counting backbone model to learn weather-aware feature representations and then refine it with a light-weight refiner. Thus, the decoder can focus on a single domain and generate high-quality density map, the sum of which is the estimated number. Since the refiner is premised on stable and consistent representation while it keeps evolving and is not stable during the representation learning stage, we divide the training into two stages to separate these two targets, namely Weather-aware Representation Learning (**WRL**) stage and Supcon-guided Representation Refining (**SRR**) stage, respectively. In the WRL stage, we use unsupervised contrastive learning to enable the encoder to learn weather-aware representations. The weights of the encoder and the decoder obtained during the WRL stage will be retained for the SRR stage. In the SRR stage, the supervised contrastive learning is utilized to refine the representations. Finally, high-quality density map can be generated to realize precise counting.

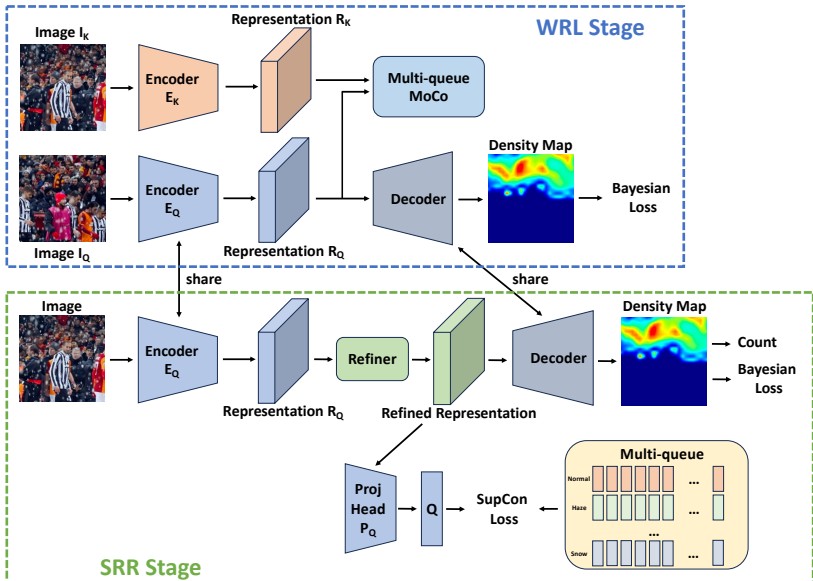

Figure 2: The architecture of MQCL. The target of the WRL stage is to learn weather-aware representations via unsupervised contrastive learning. After the WRL stage, the supervised contrastive learning is utilized in the SRR stage to realize the refinement of the representations.

## 3.2 WEATHER-AWARE REPRESENTATION LEARNING

The model architecture of the WRL stage is illustrated in the upper half of Fig. 2. The unsupervised contrastive learning method (Oord et al., 2018; Chen et al., 2020) is utilized to train the encoders and endow it with the capability to extract weather-aware representations, i.e., images with similar weather conditions correspond to similar representations, whereas those with dissimilar conditions correspond to more distant representations.

To save memory space, the contrastive learning strategy in our method is based on MoCo (He et al., 2020), which consists of an encoder $E_Q$, a momentum-updated encoder $E_K$ and a decoder. The encoder $E_Q$ extracts representation $R_Q$ from the image $I_Q$. $R_Q$ subsequently serves as the anchor in contrastive learning. To ensure that the representation simultaneously contains crowd information, the target of the decoder is set as generating a density map under the supervision of the Bayesian loss (Ma et al., 2019) according to the representation. The representation $R_K$ is extracted by the encoder $E_K$ from the image $I_K$ and will be pushed into the queues which are subsequently utilized in the calculation of the contrastive loss. Note that $I_Q$ and $I_K$ are different augmentations from the same image. Additionally, to tackle the problem of class imbalance, we propose multi-queue MoCo, the details of which will be elaborated in section 3.3. The total loss of the WRL stage is:

$$\mathcal{L}_{wrl} = \mathcal{L}_{contra} + \lambda_1 \mathcal{L}_{bayesian}, \tag{1}$$

where $\mathcal{L}_{contra}$ is the contrastive loss based on the multi-queue MoCo and $\mathcal{L}_{bayesian}$ is the Bayesian loss.

## 3.3 MULTI-QUEUE MOCO

Theoretical reasoning and experimental evidence in (Assran et al., 2022) suggest that contrastive learning has an overlooked prior-to-learn feature that enables uniform clustering of the data and it can hamper performance when training on class-imbalanced data. In the vanilla contrastive learning strategy, positive and negative samples are entirely obtained through random sampling. This strategy can work perfectly under class-balanced conditions but may struggle if the data is imbalanced due to the mismatch between the actual distribution and the model's prior.

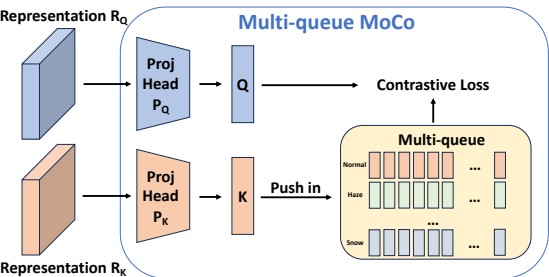

Figure 3: The architecture of multi-queue MoCo. The projection heads project the representations to 1-D vectors. In the multiple queues, each sub queue is of equal length and corresponds to one weather class.

To tackle the problem, we propose multi-queue MoCo, the architecture of which is illustrated in Fig. 3. Similar to most of the contrastive methods, each image undergoes data augmentation multiple times, and the representations originating from the same image as the anchor are treated as positive samples while those from different images are treated as negative samples. However, this strategy may lead to a situation where scenes with the same weather conditions from different images are mistakenly treated as negatives. Fortunately, research conducted by Wang & Liu (2021) indicates that contrastive learning has the tolerance to semantically similar negative samples. In light of this, we conducted extensive experiments and the results indicate that such a strategy outperforms the positive/negative partitioning strategy based on weather labels.

Both representations $R_Q$ and $R_K$ can be regarded as tensors of $\mathbb{R}^{H \times W \times C_1}$, where $H$, $W$ and $C_1$ are the height, the width and the number of channels of the representation, respectively. To avoid information loss introduced by the contrastive loss and reduce computational complexity, nonlinear projection heads are introduced after encoders to project the representations to 1-D vectors. The projection head first pools the representations to vectors of $\mathbb{R}^{C_1}$ and then project them to vectors $Q$ or $K$ of $\mathbb{R}^{C_2}$ by introducing a multi-layer perceptron. $C_2$ is the dimension of the vectors. We refer to the projection heads following encoder $E_Q$ and encoder $E_K$ as projection head $P_Q$ and projection head $P_K$, respectively. They have the same structure but do not share parameters with each other.

In contrast to MoCo, to achieve a uniform distribution of classes within vectors in the memory, we improve the original single queue to a multi-queue structure. The number of sub queues in the multiple queues is equal to the number of classes, with each sub queue having an equal length and exclusively storing vectors $Q$ that match its corresponding class. The multi-queue structure can be considered as a tensor of $\mathbb{R}^{B \times L \times C_2}$, where $B$ is the number of classes and $L$ is the length of each sub queue. Immediately when the computation of vector $Q$ is completed, it will be pushed into the corresponding sub queue. With this design, the number of samples of each weather class in the memory becomes equal, aligning perfectly with the uniform prior of the contrastive loss. Moreover, due to the limited number of images from adverse weather in the dataset, multiple samples from the same image may coexist within a sub queue. To avoid treating the above samples as negative examples, we propose to assign a unique index value to each image in the dataset and treat the samples corresponding to the same index as positive. The loss function of our multi-queue MoCo can be calculated as follows:

$$\mathcal{L}_{contra} = \sum_{i \in I} \frac{-1}{|P(i)|} \sum_{p \in P(i)} \log \frac{\exp(Q_i \cdot K_p / \tau)}{\sum_{a \in \mathbf{A}} \exp(Q_i \cdot K_a / \tau)}, \tag{2}$$

where $I$ is the batch size, $Q_i$ is the anchor vector, $P(i)$ is the set of indices of the vectors K originating from the same image with $Q_i$, $\mathbf{A}$ is the set of the indices of all of the vectors in the multiple queues and $\tau$ is the temperature.

### 3.4 SUPCON-GUIDED REPRESENTATION REFINING

After the WRL stage, we can assume that the encoder is "mature" enough to effectively extract the weather and crowd information from images. The task of the SRR stage is to train a refiner which can

convert the weather-aware representations to the normal weather domain, generating representations of the same size of $\mathbb{R}^{H \times W \times C_1}$ and enabling the decoder to focus on a single domain.

The multiple queues obtained during the WRL stage have stored a sufficient number of vectors from various weather conditions. In this paper, we propose to preserve and freeze these multiple queues and employ supervised contrastive learning to guide the refiner to convert the representation. In this stage, all the vectors in the normal-weather sub queue are treated as positive samples and those in other sub queues are treated as negatives. Since there is no longer a need to generate vectors K, the encoder $E_K$ and projection head $P_K$ are discarded. In order to maintain stable representations, the parameters of encoder $E_Q$ are fixed at this stage. The projection head $P_Q$ is preserved, fixed, and moved behind the refiner. The model structure of the SRR stage is shown in the lower half of Fig. 2. The loss function of supervised contrastive learning in this stage is calculated as follows:

$$\mathcal{L}_{supcon} = \sum_{i \in I} \frac{-1}{|\mathbf{N}|} \sum_{p \in \mathbf{N}} \log \frac{\exp(R(Q_i) \cdot K_p / \tau)}{\sum_{a \in \mathbf{A}} \exp(R(Q_i) \cdot K_a / \tau)}, \tag{3}$$

where $R(\cdot)$ is the refiner and $N$ is the set of the indices of the vectors K in the normal-weather sub queue. Similar to the WRL stage, the SRR stage continues to utilize the Bayesian loss to supervise the density map. The overall loss function $\mathcal{L}_{srr}$ for this stage is calculated as follows:

$$\mathcal{L}_{srr} = \mathcal{L}_{supcon} + \lambda_2 \mathcal{L}_{bayesian}, \tag{4}$$

where $\mathcal{L}_{supcon}$ is the supervised contrastive loss and $\mathcal{L}_{bayesian}$ is the Bayesian loss.

## 4 EXPERIMENTS AND DISCUSSIONS

### 4.1 NWPU-WEATHER DATASET

Considering the current scarcity of crowd counting datasets containing adverse weather scenarios and labels, we synthesize an adverse weather crowd counting dataset named NWPU-Weather. The specific synthetic method and the experiment setups are illustrated in the appendix.

Except for the MQCL, we also benchmark several representative counting networks, providing an overview of the state-of-the-art performance, including CSRNet (Li et al., 2018), DM-Count (Wang et al., 2020a), KDMG (Wan et al., 2020), ConvNeXt (Liu et al., 2022) and MAN (Lin et al., 2022). Note that since the model structure of AWCC-Net (Huang et al., 2023) is not fully elucidated and the code is not available at present, we do not provide the performance. Following the convention of existing works (Li et al., 2018; Lin et al., 2022), we adopt Mean Absolute Error (MAE) and Root Mean Squared Error (RMSE) as the metrics to evaluate the methods. The quantitative results of counting accuracy are listed in Table 1. Compared with the baseline ConvNeXt, MQCL exhibits a significant performance improvement under adverse weather conditions. MAE and RMSE are decreased by 13.8% and 12.6%, suggesting that the proposed representation learning and refining strategy do indeed boost the robustness of the model under adverse weather conditions. Moreover, MQCL can also improve the performance under normal conditions, decreasing MAE and RMSE by 11.1% and 7.0%, respectively. We believe that this can be attributed to the contrastive learning strategy employed in this paper, which considers augmentations from the same image as positive examples. This strategy not only assists the model in weather perception but also strengthens the model's ability to recognize different scenes. Sampling from the same image ensures that the positive examples

| Method | Normal | | Adverse | |
|---|---|---|---|---|
| | MAE ↓ | RMSE ↓ | MAE ↓ | RMSE ↓ |
| CSRNet (Li et al., 2018) (CVPR 18) | 74.8 | 200.7 | 175.3 | 730.2 |
| BL (Ma et al., 2019) (ICCV 19) | 69.9 | 248.1 | 137.1 | 339.4 |
| DM-Count (Wang et al., 2020a) (NeurIPS 20) | 80.6 | 319.1 | 153.6 | 338.2 |
| KDMG (Wan et al., 2020) (PAMI 20) | 108.0 | 318.6 | 151.9 | 328.4 |
| MAN (Lin et al., 2022) (CVPR 22) | 64.1 | 259.1 | 105.9 | 264.1 |
| ConvNeXt (Liu et al., 2022) (CVPR 22) | 69.3 | 264.0 | 108.2 | 286.3 |
| MQCL (Ours) | 61.6 (1) | 245.5 (2) | 93.3 (1) | 250.1 (1) |

Table 1: Quantitative results comparing with the state-of-the-art methods on the NWPU-Weather dataset. The numbers in parentheses represent the rankings of our method.

| Method | Normal | | Adverse | |
|---|---|---|---|---|
| | MAE ↓ | RMSE ↓ | MAE ↓ | RMSE ↓ |
| SFCN (Wang et al., 2019) (CVPR 19) | 71.4 | 225.3 | 122.8 | 606.3 |
| BL (Ma et al., 2019) (ICCV 19) | 66.2 | 200.6 | 140.1 | 675.7 |
| LSCCNN (Sam et al., 2020) (PAMI 20) | 103.8 | 399.2 | 178.0 | 744.3 |
| CG-DRCN-V (Sindagi et al., 2020) (PAMI 20) | 74.7 | 253.4 | 138.6 | 654.0 |
| CG-DRCN-R (Sindagi et al., 2020) (PAMI 20) | 64.4 | 205.9 | 120.0 | 580.8 |
| UOT (Ma et al., 2021) (AAAI 21) | 53.1 | 148.2 | 114.9 | 610.7 |
| GL (Wan et al., 2021) (CVPR 21) | 54.2 | 159.8 | 115.9 | 602.1 |
| CLTR (Liang et al., 2022) (ECCV 22) | 52.7 | 148.1 | 109.5 | 568.5 |
| MAN (Lin et al., 2022) (CVPR 22) | 46.5 | 137.9 | 105.3 | 478.4 |
| AWCC-Net (Huang et al., 2023) (ICCV 23) | 47.6 | 153.9 | 87.3 | 430.1 |
| ConvNeXt (Liu et al., 2022) (CVPR 22) | 52.7 | 154.9 | 105.1 | 561.4 |
| MQCL (Ours) | 50.5 (3) | 152.0 (4) | 96.5 (2) | 522.8 (3) |

Table 2: Quantitative results comparing with the state-of-the-art methods on the JHU-Crowd++ dataset. The numbers in parentheses represent the rankings of our method.

| Method | FLOPs | #param |
|---|---|---|
| CLTR (Liang et al., 2022) (ECCV 22) | 37.0G | 43M |
| MAN (Lin et al., 2022) (CVPR 22) | 58.2G | 31M |
| AWCC-Net (Huang et al., 2023) (ICCV 23) | 58.0G+ | 30M+ |
| ConvNeXt (Liu et al., 2022) (CVPR 22) | 27.0G | 29M |
| MQCL (Ours) | 31.2G | 32M |

Table 3: Comparison of computational complexity and the number of parameters. The computational complexity is measured by FLOPs when inferencing images with the size of $384 \times 384$.

not only share the same weather conditions but also possess similar scene characteristics. Compared with the previously best method MAN, MQCL has also achieved significant performance improvement. The MAE/RMSE under normal and adverse weather conditions have reduced by 3.9%/5.2% and 11.9%/5.3%, respectively.

## 4.2 JHU-CROWD++ DATASET

As shown in Table 2, despite the challenges posed by the diverse scenes, complex and variable weather conditions and weather class imbalance in the JHU-Crowd++ dataset, MQCL achieves an improvement of 8.6% in MAE and 6.9% in RMSE under adverse weather conditions compared to the baseline ConvNeXt. This indicates that MQCL not only performs well on synthetic datasets but also effectively boosts the model's robustness in real-world datasets. However, while MQCL achieved significant improvements over the baseline model ConvNeXt, its performance still slightly lags behind the current state-of-the-art algorithms. We summarize the reasons as follows: 1) as shown in Table 2, all methods outperforming MQCL, including CLRT, MAN, and AWCC-Net, have introduced Transformer-based modules, the computational complexity of which is significantly higher than MQCL. The MAE of AWCC-Net under adverse weather conditions is 9.5% lower than our method, but it incurs over 85.9% more FLOPs; 2) MQCL focuses on the design of learning strategies. It only introduces a lightweight refiner module without significantly altering the architecture of the backbone network.

## 4.3 DISCUSSIONS ABOUT CONTRASTIVE LEARNING

**The significance of contrastive learning.** As discussed earlier, the prerequisite for addressing the multi-domain learning problem is that the model can perceive different domains. If the encoder fails to extract distinctive weather information, the refiner will lack the target for conversion and struggle to fulfill its intended purpose. As shown in Table 4, if contrastive learning is not employed in the first stage to enable the model to learn weather information with discriminative characteristics, and the refiner module is directly introduced afterwards, the performance of the model actually decreases significantly under both normal and adverse weather conditions. We believe this is because the refiner module, trained from scratch without clear targets, may disrupt the crowd information extracted by the original encoder.

| Method | Normal | | Adverse | |
|---|---|---|---|---|
| | MAE ↓ | RMSE ↓ | MAE ↓ | RMSE ↓ |
| Encoder and decoder only | 52.7 | 154.9 | 105.1 | 561.4 |
| Refine w/o CL | 56.2 | 163.1 | 107.1 | 585.6 |
| MQCL (Ours) | **50.5** | **152.0** | **96.5** | **522.8** |

Table 4: The significance of contrastive learning in our method. "Refine w/o CL" adopts a similar approach to MQCL, except that $\mathcal{L}_{contra}$ and $\mathcal{L}_{supcon}$ are removed from the loss functions, respectively.

| Method | Normal | | Adverse | |
|---|---|---|---|---|
| | MAE ↓ | RMSE ↓ | MAE ↓ | RMSE ↓ |
| Strategy 1 | 51.4 | 153.0 | 99.7 | 540.4 |
| Strategy 2 (Ours) | **50.5** | **152.0** | **96.5** | **522.8** |

Table 5: Performance comparison of the two optional strategies on the JHU-Crowd++ dataset. Strategy 1 treats samples with the same weather label as positive examples and strategy 2 treats samples originating from the same image as positive examples.

**Strategy of positive/negative selection.** As mentioned above, there are two optional strategies in the WRL stage: 1) treating samples with the same weather label as positive examples; 2) treating samples originating from the same image as positive examples; We conduct experiments on the JHU-Crowd++ dataset to compare these two strategies. The model performance after the WRL stage using these two strategies is shown in Table 5. Strategy 2 outperforms Strategy 1 in both normal and adverse conditions. We attribute this to the following reasons: 1) even if two samples share the same weather label, their weather conditions may still vary significantly. The practice of minimizing all the representations with the same label is not in line with the target of us; 2) samples from the same image not only share the same weather conditions but also the same scene. Strategy 2 has the potential to enhance the model's scene recognition capabilities.

**The storage strategy for vectors K.** There are three optional storage strategies for vectors K: 1) memory bank (Wu et al., 2018); 2) single-queue MoCo (He et al., 2020); 3) multi-queue MoCo (ours). Extensive experiments are conducted on the JHU-Crowd++ dataset to compare the above strategies. The t-SNE (Van der Maaten & Hinton, 2008) visualization of the vectors Q after the WRL stage on the JHU-Crowd++ dataset is shown in Fig. 4. The memory bank strategy suffers from a severe lack of discriminative capacity. The representations learned by the single-queue strategy are also not discriminative enough, especially near the rain weather representations in Fig. 4(b). The multi-queue strategy does not suffer from the aforementioned issues. The performance comparison shown in Table 6 also demonstrates that the proposed multi-queue MoCo can effectively address the class imbalance problem. We attribute the phenomenon to the following reasons: 1) while the memory bank can store a large number of samples with minimal memory consumption, it does not employ a stable strategy to update the encoder, and the sample update frequency is too low, resulting in poor sample consistency; 2) although single-queue MoCo addresses the issue of poor sample consistency by introducing a queue and momentum update strategy, the class imbalance problem in the dataset leads to inconsistencies between the data distribution in the queue and the uniform

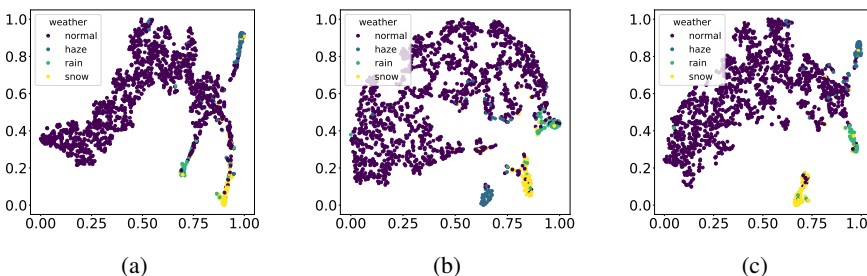

(a)        (b)        (c)

Figure 4: The t-SNE (Van der Maaten & Hinton, 2008) visualization of the vectors Q after the WRL stage on the JHU-Crowd++ dataset using memory bank (a), single-queue MoCo (b) and multi-queue MoCo (c), respectively.

| Method | Normal | | Adverse | |
|---|---|---|---|---|
| | MAE ↓ | RMSE ↓ | MAE ↓ | RMSE ↓ |
| Memory Bank | 53.7 | 157.6 | 104.9 | 531.1 |
| Single-queue MoCo | 54.6 | 166.5 | 107.9 | 589.7 |
| Multi-queue MoCo (Ours) | **50.5** | **152.0** | **96.5** | **522.8** |

Table 6: Performance comparison of the three storage strategies for vectors K on the JHU-Crowd++ dataset.

| Method | Normal | | Adverse | |
|---|---|---|---|---|
| | MAE ↓ | RMSE ↓ | MAE ↓ | RMSE ↓ |
| Encoder and decoder only | 52.7 | 154.9 | 105.1 | 561.4 |
| + Multi-queue MoCo | 51.2 | 152.6 | 97.2 | 524.6 |
| + Refiner | 50.5 | 152.0 | 96.5 | 522.8 |

Table 7: Ablation study on the JHU-Crowd++ dataset.

distribution prior; 3) multi-queue MoCo not only retains the advantages of high sample consistency but also greatly alleviates the problem of inconsistency between data distribution and the uniform prior. Thus, it achieves the best performance.

## 4.4 ABLATION STUDY

Ablation studies are performed on the JHU-Crowd++ dataset and the quantitative results are shown in Table 7. We start with the baseline of the end-to-end model, i.e., only the encoder and decoder. First, the effectiveness of multi-queue MoCo is tested. An improvement of 2.8%/1.5% and 7.5%/6.6% in MAE/RMSE under normal and adverse weather is achieved compared to the baseline. From this, it can be seen that most of the performance improvements under adverse weather conditions come from the representation learning in the WRL stage. This verifies the effectiveness of the proposed multi-queue MoCo for enhancing robustness under adverse weather conditions. Additionally, the performance improvement under normal weather conditions corroborates the earlier analysis that the strategy that treats different augmentations from the same image as positive samples can aid in strengthening the scene recognition capabilities of the model. Subsequently, the refiner is added, and the best performance is achieved, with a reduction of 1.4%/0.4% in MAE/RMSE under normal weather conditions and 0.7%/0.3% under adverse weather conditions, respectively. This demonstrates that, under the guidance of supervised contrastive learning, the refiner is capable of converting adverse weather representations to the normal domain, enabling the decoder to focus on a single domain, resulting in performance improvements under both normal and adverse weather conditions.

## 5 CONCLUSION AND LIMITATION

In this paper, we propose a contrastive learning-based method called MQCL to tackle the problem of class-imbalanced adverse weather crowd counting and synthesize a new adverse weather crowd counting dataset. To address the dual challenges of image degradation and class imbalance, the multi-queue MoCo is employed to enable the model to learn weather-aware representations. Furthermore, supervised contrastive learning is utilized to guide the refiner on representation conversion. Extensive experiments are conducted to compare and choose the strategy of contrastive learning. We believe that this is not only applicable to the crowd counting task but also holds significant reference value for other domains.

MQCL has achieved significant performance improvements compared to the baseline. However, there are still some limitations. For example, we focus on the learning strategy and maintaining the model's lightweight nature, but its performance still falls slightly behind the state-of-the-art models based on Transformers. Secondly, we only utilized synthetic methods to construct the dataset, leading to a certain domain gap between real-world scenes.

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

## A  APPENDIX INTRODUCTION

In this appendix, we list more details of our paper: 1) The experiment setup, including the implementation details, the method of data augmentation and the hyper-parameter settings; 2) The introduction to the datasets, including the specific synthetic method for the NWPU-Weather dataset and detailed information about the JHU-Crowd++ dataset (Sindagi et al., 2020).

## B  EXPERIMENT SETUP

The ConvNeXt (Liu et al., 2022) is employed as our backbone model and we use the ConvNeXt-T version for simplicity. The structure of ConvNeXt-T is: C = (96, 192, 384, 768), B = (3, 3, 9, 3), where C stands for the number of channels and B is the number of ConvNeXt blocks in each stage. Following Ling et al. (2023), we replace the linear layer at the end of the ConvNeXt by an upsampling block to keep the downsampling rate at 8. The encoder $E_Q$ and encoder $E_K$ both have the same structure as the first three stages of ConvNeXt and the decoder corresponds to the final stage. The pre-trained weights of ImageNet-22K (Russakovsky et al., 2015) are loaded as the initial parameters. Two-layer MLPs are employed in the projection heads, the output dimension of which is 2048 and 128. The refiner consists of three ConvNeXt blocks with input and output dimensions of 768, initialized with random parameters.

As mentioned above, each image in a batch undergoes augmentation twice. Specifically, we random crop the image with a size of $256 \times 256$, and horizontal flipping is applied for a probability of 50%. In the multi-queue structure, the number of the sub queues is equal to the number of weather classes, and the length of each sub queue is set to 1024. The AdamW optimizer (Loshchilov & Hutter, 2017) is adopted both in the WRL and SRR stage, the learning rate is scheduled by a cosine annealing strategy and the initial learning rate is $10^{-4}$. The weight decay is set to $10^{-3}$ and the batch size is 16. $\lambda_1$ and $\lambda_2$ in the loss function are both set to 10 and the temperature $\tau$ is set to 0.05.

## C  INTRODUCTION TO THE DATASETS

### C.1  NWPU-WEATHER DATASET

Considering the current scarcity of crowd counting datasets containing adverse weather scenarios and labels, we synthesize an adverse weather crowd counting dataset containing hazy and rainy scenes based on the NWPU-Crowd dataset (Wang et al., 2020b), namely NWPU-Weather. Since the test set of the NWPU-Crowd dataset is not publicly available, we extract part of the original training set as the test set. The first 1-2609 images from the original dataset are used as the training set, the images from 2610 to 3109 are used as the test set, and the validation set still consists of images from 3110 to 3609.

Consistent with the JHU-Crowd++ dataset, our NWPU-Weather dataset keeps the imbalance of weather types, the hazy and rainy scenes are synthesized by a probability of 5%, respectively. The distribution of the weather conditions in the dataset is shown in Table 8. We follow the approach outlined in (Li et al., 2019) to synthesize the weather scenarios. The intensity, density, and angle of the rain are set as random values. The depth maps required during the haze synthesis process are estimated by ZoeDepth (Bhat et al., 2023) and the intensity of the haze is also random.

| Stage | Normal | Haze | Rain |
|-------|--------|------|------|
| Train | 2365   | 120  | 124  |
| Val   | 446    | 31   | 23   |
| Test  | 442    | 28   | 30   |
| Total | 3253   | 179  | 177  |

Table 8: The distribution of the number of images under different weather conditions in the NWPU-Weather dataset.

## C.2   JHU-CROWD++ DATASET

There are 4372 images and 1.51 million labels contained in the JHU-Crowd++ dataset (Sindagi et al., 2020). Out of these, 2272 images were used for training, 500 images for validation, and the remaining 1600 images for testing. The advantage of JHU-Crowd++ is its inclusion of diverse scenes and environmental conditions, such as rain, snow and haze. It also provides weather condition labels for each image. Due to the rarity of adverse weather, the weather classes in JHU-Crowd++ are imbalanced. As is shown in Fig. 1(c), the number of images under rain, snow, and haze conditions accounts for only 3%, 5%, and 4% of the total dataset, respectively.

