# OpenReview forum: "Boosting Adverse Weather Crowd Counting via Multi-queue Contrastive Learning"
_ICLR.cc/2024/Conference — Submitted to ICLR 2024_

### Official Review · Reviewer_nR6L · 2023-10-31

**Soundness:** 2 fair
**Presentation:** 2 fair
**Contribution:** 2 fair
**Rating:** 5
**Confidence:** 5

**Summary:**

This paper proposes a contrastive learning-based method to deal with class imbalance for different weathers in crowd counting. A synthetic dataset with different weather augmentation is proposed based on. NWPU-Crowd. The unsupervised contrastive learning with multi-queue MoCo is proposed to learn better features for imbalanced classes with adverse weather. Then, a supervised contrastive learning is proposed to refine the learned feature.

**Strengths:**

The proposed method achieves good improvement based on adverse weather.

**Weaknesses:**

1. Section 3.2 presents weather-aware representation learning. However, it is unclear why contrastive learning can be used to learn weather-aware representation learning.
2. The proposed method is not compared with other class imbalance strategies which is not convincing.
3. The presentation is unclear. For example, in the caption in Figure 2, WRL is an unsupervised contrastive learning method while in Section 2, a supervised loss is used during training, which is confusing.

**Questions:**

1. The method is a generic method to deal with data imbalance. Is it effective for other applications with data imbalance issues?
2. Is multi-queue MoCo part of WRL or SRR? From Figure 2, multi-queue is part of SRR, but as described in the main text, it’s part of WRL.
3. How is the generalization ability of the model trained on synthetic data? If the model trained on synthetic data can't be generalized to the real dataset, the contribution of the synthetic dataset is limited.

---

### Official Review · Reviewer_evYm · 2023-11-01

**Soundness:** 2 fair
**Presentation:** 3 good
**Contribution:** 3 good
**Rating:** 5
**Confidence:** 4

**Summary:**

This paper presents a two-stage method to address the adverse weather problem in crowd counting. In the first stage, a multi-queue MoCo is established to capture crowd features in different weather conditions in an unsupervised manner. In the second stage, a refining module is placed before the density decoder to transform crowds in adverse weather conditions into normal conditions, resulting in improved counting performance in adverse weather conditions.

**Strengths:**

- An unsupervised way is designed based on MoCo to extract weather condition prosperity in crowd image.
- A refiner is learned to transform images in abnormal weather into normal conditions for better prediction.
- The NWPU-Weather dataset is extracted from the NWPU-Crowd dataset for this new task.

**Weaknesses:**

- No visualization is provided for the unsupervised results. Whether the model works as expected is unclear.
- How the multi-queue MoCo is trained is a little unclear. see questions.

**Questions:**

1. Could the authors provide some images belonging to different queues in the multi-queue MoCo?
2. What is the strategy for assigning Q to the corresponding sub-queue? How is the relationship between the current Q and different sub-queues modeled?
3. What is the difference between Eq. (2) and the vanilla MoCo? How does the inclusion of different sub-queues affect this loss function?
4.  can it be ensured that different sub-queues contain crowds from different weather conditions? Since this is an unsupervised approach, it can be assumed that images with similar properties could be assigned to the same category. The question is whether the properties learned from the multi-queue MoCo represent weather information.
5. The SRR stage requires a label for the normal condition. How is this label defined based on the sub-queues?
6. The GCC[1] dataset contains weather information (7 weather types) for each image. Perhaps the authors could analyze their unsupervised WRL on this dataset. Specifically, they can directly apply their method to this dataset and compare the unsupervised results with the provided ground truth to demonstrate whether the model performs as expected.

---

### Official Review · Reviewer_WLVr · 2023-11-06

**Soundness:** 2 fair
**Presentation:** 2 fair
**Contribution:** 2 fair
**Rating:** 3
**Confidence:** 5

**Summary:**

This paper focuses on crowd counting under extreme and adverse weather conditions. Specifically, the authors propose a two-stage framework, where the first stage introduces a multi-queue MoCo contrastive learning strategy to tackle the problem of weather class imbalance, and the second stage adopts the supervised contrastive loss to guide the refinement process. Extensive experiments demonstrate the effectiveness of the proposed method.

**Strengths:**

This paper is well written and the motivation is clear. The proposed method can effectively improve the robustness of the adverse weather conditions.

**Weaknesses:**

I am not sure of the importance of this task. Since this topic is too small, it might not be interesting for the researcher in computer vision, even in crowd counting. Considering the high bar of ICLR, I do not think this task can match the ICLR.

What about simply using the existing image restoration methods to make pre-processing? I think the authors should have a discussion.

The experiments are not convincing. First, as shown in Tab.2, compared with previous methods, the performance of adverse samples is improved by sacrificing the performance of a portion of normal samples. Second, the authors should report the average performance of the whole dataset, not only normal and adverse. Third, compared with previous SOTA, the proposed method does not have significant performance gain.

What about the cross-domain performance? e.g., training on JHU-Crowd and testing on NWPU-Crowd weather.

**Questions:**

see weakness

---

### Meta-Review · Area_Chair_zTfR · 2023-12-06

**Metareview:**

The paper was reviewed by 3 experts.  The major concerns were the narrow scope of the problem, unconvincing experiments, and lack of simple baselines (pre-processing with image restoration), unclear motivation/justification of the proposed method.
There was no response submitted. The AC agrees with the comments, and thus recommends reject.

**Justification For Why Not Higher Score:**

The concerns are valid, and there was no author response.

**Justification For Why Not Lower Score:**

n/a

---

### Decision · Program_Chairs · 2024-01-16

Reject